# *Limosilactobacillus reuteri* HCS02-001 Attenuates Hyperuricemia through Gut Microbiota-Dependent Regulation of Uric Acid Biosynthesis and Excretion

**DOI:** 10.3390/microorganisms12040637

**Published:** 2024-03-22

**Authors:** Akbar Hussain, Binqi Rui, Hayan Ullah, Panpan Dai, Kabir Ahmad, Jieli Yuan, Yinhui Liu, Ming Li

**Affiliations:** 1College of Basic Medical Science, Dalian Medical University, Dalian 116041, China; akbarhussain135@gmail.com (A.H.); ruibinqi7270@163.com (B.R.); hayan.khan12@gmail.com (H.U.); zgwst@126.com (J.Y.); 2Department of Physiology, Dalian Medical University, Dalian 116041, China; kbrahmad4@gmail.com

**Keywords:** hyperuricemia, probiotic, inflammation, uric acid transporter, gut microbiota, transcriptome

## Abstract

Hyperuricemia is a prevalent metabolic disorder that arises from abnormal purine metabolism and reduced excretion of uric acid (UA). The gut microbiota plays a significant role in the biosynthesis and excretion of UA. Probiotics capable of purine degradation possess the potential to prevent hyperuricemia. Our study aimed to screen probiotics in areas with abundant dairy products and longevity populations in China, which could attenuate the level of UA and explore the underlying mechanism. In this study, twenty-three lactic acid bacteria isolated from healthy Chinese infant feces and traditional fermented foods such as hurood and lump milk were evaluated for the ability to tolerance acid, bile, artificial gastric juice, and artificial intestinal juice to determine the potential of the candidate strains as probiotics. Eight strains were identified as possessing superior tolerance to simulated intestinal conditions and were further analyzed by high-performance liquid chromatography (HPLC), revealing that *Limosilactobacillus reuteri* HCS02-001 (Lact-1) and *Lacticaseibacillus paracasei* HCS17-040 (Lact-2) possess the most potent ability to degrade purine nucleosides. The effect of Lact-1 and Lact-2 on hyperuricemia was evaluated by intervening with them in the potassium oxonate and adenine-induced hyperuricemia Balb/c mice model in vivo. Our results showed that the level of serum UA in hyperuricemic mice can be efficiently reduced via the oral administration of Lact-1 (*p* < 0.05). It significantly inhibited the levels of liver inflammatory cytokines and hepatic xanthine oxidase through a TLR4/MyD88/NF-κB pathway across the gut–liver axis. Furthermore, UA transporters ABCG2 and SLC2A9 were substantially upregulated by the intervention of this probiotic. Fecal ATP levels were significantly induced, while fecal xanthine dehydrogenase and allantoinase levels were increased following probiotics. RNA sequencing of HT-29 cells line treated with Lact-1 and its metabolites demonstrated significant regulation of pathways related to hyperuricemia. In summary, these findings demonstrate that *Limosilactobacillus reuteri* HCS02-001 possesses a capacity to ameliorate hyperuricemia by inhibiting UA biosynthesis via enhancing gastrointestinal barrier functions and promoting UA removal through the upregulation of urate transporters, thereby providing a basis for the probiotic formulation by targeting the gut microbiota.

## 1. Introduction

Hyperuricemia is a prevalent metabolic disorder that arises from abnormal purine metabolism and/or reduced excretion of UA. Individuals whose serum UA (SUA) levels exceed 420 μmol/L in males and 360 μmol/L in females may be clinically identified as having hyperuricemia. Over the past few decades, there has been a global increase in the prevalence of hyperuricemia and gout. The prevalence of hyperuricemia among Chinese adults was 8.4%, indicating that roughly 92.9 million persons in China were affected by hyperuricemia [1]. In addition, the presence of asymptomatic hyperuricemia, a condition that has attracted relatively less attention from both researchers and the general public, has been found to be associated with a higher incidence of many disorders such as hypertension, acute and chronic renal disease, metabolic syndrome, obesity, type 2 diabetes mellitus, and non-alcoholic fatty liver disease [2,3]. Furthermore, there is a high prevalence of hyperuricemia among individuals suffering from severe chronic systolic heart failure, and it has been observed that hyperuricemia is associated with increased rates of hospitalization and mortality [4]. Globally, the incidence rate of hyperuricemia has increased by 20% [5,6,7]. Hypertension, hyperlipidemia, and hyperglycemia have been identified as the three primary risk factors, along with hyperuricemia being recognized as the fourth-highest risk factor [8].

The current treatment regimen for hyperuricemia includes dietary modifications, physical activity, pharmacotherapy, and biotherapeutic interventions. These approaches seek to either reduce the generation of UA or enhance its excretion [9]. Dietary intervention aimed at lowering the consumption of high-purine food is a cost-effective approach with no or negligible adverse effects for a significant number of patients diagnosed with hyperuricemia. However, it is worth noting that patient adherence to this strategy is generally suboptimal [10,11]. Allopurinol, a pharmaceutical agent known for its ability to reduce levels of hyperuricemia, functions by competitively inhibiting the xanthine oxidase (XO) enzyme, thus impeding the synthesis of UA. Uricosuric agents, such as benzbromarone, sulphinpyrazone, and probenecid, have the ability to increase the excretion of UA from the body. Despite the advancements made in enhancing the safety of XO inhibitors and uricosuric medicines, their clinical utility is frequently constrained due to the occurrence of adverse effects, including headaches, diarrhea, rashes, severe allergic responses, and nephrotoxicity [12,13]. Hence, it is imperative to identify an efficacious strategy that promotes improved patient compliance and minimizes adverse effects.

Gut microbiota is closely linked to many diseases and plays a crucial role in the metabolism of purines and UA. Gout patients’ gut microbiomes are distinct, showing a marked decline in bacteria that express the uricase gene. Additionally, the changed gut microbiota affects the expression of UA transporters in the intestine, including ABCG2 and GLUT9, as well as the ongoing inflammatory response, both of which aid in the development of hyperuricemia [14,15,16]. Therefore, it is possible to utilize probiotics to reduce hyperuricemia and the inflammatory damage brought on by urate in urine. Numerous strains of *Lactobacillus* have been identified by researchers to lower SUA levels. For instance, *L. fermentum* JL-3 from the traditional Chinese cuisine “Jiangshui” degraded UA in vitro and reduced hyperuricemia in mice [17]. *L. gasseri* PA-3 can directly use AMP, IMP, and GMP for growth and reduce rat purine absorption [18,19]. *L. rhamnosus* 1155 and *L. fermentum* 2644 were isolated from traditional fermented products and showed capability against purine nucleoside and the prevention of hyperuricemia [20]. *L. Paracasei* extracted from traditional pickles degraded nucleosides in vitro and UA in vivo, alleviating kidney damage in a rat model of chronic hyperuricemia [21]; other strains of *L. paracasei* X11 restored intestinal microbiota and reduced UA in hyperuricemia mice [22]. The study reported that potential probiotics *L. paracasei* MJM60396 can prevent hyperuricemia by absorbing purines, inhibiting xanthine oxidase, increasing UA excretion, and modulating urate transporters in mice [23]. *L. reuteri* strain TSR332 and *L. fermentum* strain TSF331 effectively reduced hyperuricemia and stabilized UA levels in rats [24]. However, these studies only focused on the degradation of purines and nucleosides in the screening of strains, without paying attention to the gastrointestinal tolerance of the strains and whether they have probiotic potential.

Inner Mongolia is the largest milk source region in China. The region has developed animal husbandry, with beef and mutton as the main food. Despite a diet rich in purines, the incidence of hyperuricemia and gout was not significantly higher than elsewhere [25]. It was reported that fermented dairy products have the effect of reducing SUA [26], and traditional fermented dairy products are also the daily diet of people in Inner Mongolia. Therefore, we speculated that probiotics with uric-acid-lowering effects may be present in traditional fermented dairy products in the region. The Bama people in Guangxi, China, are one of the longest-living populations in the world [27]. Whether there are probiotics that can potentially reduce SUA in the gut microbiota of the longevity population area is also a research direction that we are interested in.

In our study, probiotics were screened from the feces of healthy infants, and traditional fermented dairy products such as milk curd and lump milk were sourced from diverse regions in Inner Mongolia and Guangxi. Subsequently, gastrointestinal tolerance was detected to ensure the potential to be probiotics. The purine and nucleoside decomposing capacity of probiotics was detected in vitro; based on their significant decomposition activities, Lact-1 and Lact-2 were selected for further study. We used Lact-1 and Lact-2 against the potassium oxonate- and adenine-induced hyperuricemia mouse model in vivo to verify the UA-lowing activity of them. The underlying mechanism was explored via the transcriptome sequencing of Lact-1 on HT-29 cells in vitro. This study lays the foundation for the development of probiotics to prevent or ameliorate hyperuricemia and provide a novel prevention and amelioration strategy for hyperuricemia.

## 2. Materials and Methods

### 2.1. Probiotic Screening

#### 2.1.1. Strain Cultivation

In total, 23 lactic acid bacteria strains were isolated from the feces of healthy Chinese infants and traditional fermented foods such as milk curd and lump milk were sourced from diverse regions in Inner Mongolia and Guangxi, China, (Table 1) and stored at −80 °C. The sample collection was approved by the ethics committees of Dalian Women and Children Medical Center (Group), Dalian, China (No. 2022018). In MRS liquid medium (QingDao Hopebio Technology, Qingdao, China), 2% (*v*/*v*) strains were inoculated and cultured at 37 °C for 24 h. Optical density (OD) was measured at 600 nm to determine the growth state for use.

#### 2.1.2. Tolerance to Acid, Bile, Artificial Gastric and Intestinal Fluids

The 8 isolated strains solution was inoculated into MRS solid medium and cultured at 37 °C for 48 h. A single colony with consistent growth was selected and inoculated into 5 mL liquid MRS Medium. After culturing at 37 °C for 20 h, the 5 mL cultured bacterial solution was inoculated into 100 mL liquid MRS medium and cultured at 37 °C for 20 h.

The bacteria were generated 3 times, inoculated in the MRS medium (pH 3.0), and cultured at 37 °C for an acid resistance test. The samples were periodically withdrawn at 0 h and 3 h, and a 1 mL sample was then transferred into 9 mL of sterile saline solution by serial dilutions. A 0.1 mL portion of the suspension was cultured on an MRS agar plate at 37 °C for 24 h. Strains sampled at 0 h were used as a control. CFUs were counted, and the survival rate in acid was determined. Likewise, the bacteria were inoculated in the MRS (0.3% bile salt, pH 8.0) and cultured at 37 °C for a bile tolerance test. A sample volume of 1 mL was taken at 0 h and 4 h, transferred into 9 mL of sterile saline solution, and mixed. The survival rate in the bile was then detected using the same manipulation. 

Simulated gastric juices were prepared with 3 g/L pepsin (1:10,000, Sigma, St. Louis, MO, USA) in phosphate-buffered saline solution (PBS; pH 3.0) and filtered with a 0.22-µm filter before use. Ten mL bacterial suspension was centrifuged (7220 rpm, 10 min, 4 °C) to obtain bacterial mud and washed with PBS twice. Each washed cell suspensions (1 mL) of 8 strains were mixed with 9 mL of the simulated gastric juice and cultured at 37 °C. The samples were periodically withdrawn at 0 h and 3 h, and the survival rate in gastric juices was determined. Simulated intestinal juices were prepared using 1 g/L trypsin (1:250, Sigma, St. Louis, MO, USA) in PBS (pH 8.0) and filtered with a 0.22-µm filter before use. Each washed cell suspensions (1 mL) of the 8 strains were mixed with 9 mL of sterile intestine juice and cultured at 37 °C. A sample volume of 1 mL was taken at 0 h and 4 h, transferred into 9 mL of sterile saline solution, and mixed. The survival rate in the intestinal juice was then detected using the same manipulation.
Survival rate of tested strains (%) = C_3h or 4h_/C_0h_ × 100%.

#### 2.1.3. Determination of Nucleoside Degradation Activity In Vitro

Inosine and guanosine were determined by high-performance liquid chromatography (HPLC). The solutions were injected into an HPLC device (LC-20A, Shinadzu Corporation, Kyoto, Japan). The isocratic elution was performed using a NaClO_4_-H_3_PO_4_ solution at a flow rate of 1 mL/min. The contents of inosine and guanosine were identified at 254 nm and quantified via the interpolation of calibration curves. 

The activated strains were inoculated with MRS liquid medium and cultured at 37°C for 24 h. One mL of culture medium (1.2 × 10^9^ CFU) was centrifuged at 4000 rpm at 4 °C for 10 min. The bacteria were collected, washed with 1 mL sterile normal saline (0.9% NaCl), and repeated 3 times. In total, 750 µL of inosine-guanyl culture solution (0.337 g/L inosine; 0.357 g/L guanyl; 0.1 mol/L K_3_PO_4_ solution; pH = 7.0) was added and oscillated for 1 h at 120 rpm at 37 °C. The above steps were repeated, and the obtained mixture was then oscillated for 2 h at 120 rpm at 37 °C. At the same time, the culture medium of uninoculated strain was set as the blank control. The mixture obtained after degradation for 1 h or 2 h was then centrifuged at 4000 rpm for 10 min. Next, 270 µL of supernatant was taken, 30 µL of 0.1 mol/L HClO_4_ solution was added, and 20 µL of solution was taken for high-performance liquid phase (HILIC-UV) analysis; this process was repeated 3 times.

### 2.2. Animals and Study Design

Specific pathogen-free (SPF) Balb/c mice (male; 12 weeks of age) were purchased from Liaoning Changsheng Biotechnology Co., Ltd. (Benxi, China) (Animal Qualification Certificate Number: SCXK(Liao)2020---0001. Mice were housed at 18–24 °C in 12 h light/dark cycles, and the relative humidity of the animal room was 50–70%. The mice were acclimatized for 1 week before use. All animals were maintained in individual cages, and the total experimental duration was 4 weeks. The experiment was conducted in accordance with the animal management regulations of the Ministry of Science and Technology of China and approved by the Experimental Animal Ethics Committee of the Dalian Medical University, China (Permission number: SYXK [Liao] 2018–0002). The male Balb/c mice were randomly assigned to 4 groups: control group (CON), hyperuricemia group (MOD), *Limosilactobacillus reuteri* HCS02-001 (Lact-1), and *Lacticaseibacillus paracasei* HCS17-040 (Lact-2) (*n* = 4). The establishment of the hyperuricemia model is shown in Figure 1. The hyperuricemia model group, i.e., the Lact-1 and Lact-2 group, was developed via the oral administration of potassium oxonate 200 mg/kg and adenine 50 mg/kg (Sigma-Aldrich Co., St. Louis, MO, USA). The control and hyperuricemia groups were given 200 μL of normal saline (0.9% NaCl), whereas the treated groups were orally administrated 200 μL of Lact-1 and Lact-2 (1 × 10^9^ CFU/mL). After 2 h, except for the normal group, the other groups were given the modeling drug for 21 days. All groups were fed a normal diet and water. Feces were collected on 7th, 14th, and 21st days from day 0 of the treatment. Mice were sacrificed on the 21st day, and the blood, hepatic, and intestinal tissues were collected and stored at −80 °C until use. When blood was collected, isoflurane was used for general anesthesia in mice. After blood collection, the mice were sacrificed by carbon dioxide euthanasia.

### 2.3. Molecular Biology Experiment Design

#### 2.3.1. Biochemical Parameters Analysis

The biochemical parameters such as UA and creatinine were analyzed using respective detection kits (Nanjing Jiancheng Bioengineering Institute, Nanjing, China). Lipopolysaccharide (LPS), Interleukin 1β (IL-1β), Interleukin-6 (IL-6), and Interleukin-18 (IL-18) activities were measured using the enzyme-linked immunosorbent assay (ELISA) kit (Lengton, Shanghai, China) by following the manufacturer’s guidelines and instructions. UA and creatinine were measured from blood serum, while ATP (Solarbio BC0300^®^) and Amino Acid (Solarbio BC1575^®^, Beijing, China) were measured from feces using a colorimetric assay kit by following the manufacturer’s instructions (Thermo Fisher, Waltham, MA, USA).

#### 2.3.2. RNA Isolation, cDNA Synthesis, and Quantitative Real-Time PCR (qPCR)

A total of 50 mg tissue samples were homogenized, and 1 mL of Trizol reagent was added (Invitrogen; Carlsbad, CA, USA). Following the manufacturer’s instructions, total RNA was extracted from the sample tissue. The complementary DNA was subsequently transcribed with the commercially available HiScript II Q RT SuperMix kit (Vazyme Biotech Co., Ltd., Nanjing, China). For RT-qPCR, the primers of XOD, TLR4, MyD88, NF-κB Claudin-1, Claudin-4, Occludin, ZO-1, ABCG2, SLC2A9, XHD, Allantoinase, and β-actin were used. Primers used in this study are shown in Table 2.

#### 2.3.3. Western Blotting

The protein (ABCG2, SLC2A9) extraction from tissue samples was documented in the literature [28]. SDS-polyacrylamide gel electrophoresis (SDS-PAGE) was used for the purpose of protein isolation, followed by the subsequent transfer of the isolated proteins onto a PVDF membrane, and then the membrane was blocked using 5% skim milk with TBST. After that, the membrane was washed three times with TBST. The target band was subjected to molecular weight-based separation and afterward treated overnight at 4 °C using primary antibodies (ABCG2, SLC2A9, and GAPDH). On a subsequent day, the blots underwent a triple wash with TBST and were subsequently subjected to incubation with the secondary antibody for 1 h at room temperature; then, the blots were washed three times with TBST and were subsequently developed using Enhanced Chemiluminescence. The visualization of protein bands was carried out using the ChemiDoc MP Imaging System (Bio-Rad, Hercules, CA, USA). 

### 2.4. Genomic DNA Extraction and 16S rRNA Sequencing and Data Analysis

The QIAamp DNA stool micro kit (Qiagen, Düsseldorf, Germany) was used to extract the meta-genomic DNA from mouse feces. The quality and quantity of the DNA were assessed using a NanoDrop 2000 spectrophotometer (Thermo Scientific, Waltham, MA, USA). The amplification of the V4 region of the microbial 16S rRNA gene was performed using the forward primer 515F (5′- GTGCCAGCMGCCGCGGTAA-3′) and the reverse primer 806R (5′-GGACTACHVGGGTWTCTAAT-3′). Subsequently, the amplified DNA fragments were sequenced using the Illumina NovaSeq6000 platform (GUHE Info technology Co., Ltd., Hangzhou, China). The sequence data processing was performed using the QIIME software version 1.9 pipeline, as previously described [29]. The analysis involved the utilization of Vsearch V2.4.4 to perform paired-end read alignment, operational taxonomic unit (OTU) clustering, detection of chimera clusters, and dereplication using the default parameters. The taxonomic classification was determined for each dataset by utilizing the Green genes database.

### 2.5. Cell Lines Culturing

The colon epithelial cell lines (HT-29) were acquired from the Shanghai Cell Bank. The cells were grown in Mccoy’5A medium supplemented with 10% fetal bovine serum (HyClone; GE Healthcare life sciences, Logan, UT, USA) and an antibacterial agent (100 μg/mL streptomycin, 100 units/mL penicillin) purchased from KeyGen Biotech (Shanghai, China).

### 2.6. Transcriptomic Analysis of Metabolites- and Probiotics-Treated HT-29 by RNA Sequencing

For RNA sequencing, total RNA was collected from HT-29 cells treated for 12 h with Lact-1 and its metabolites, and experiments were performed in triplicate. Total RNA was extracted using the RNA Nano 6000 Assay Kit (Agilent Technologies, Santa Clara, CA, USA), and an Agilent Bioanalyzer 2100 system was used to check the RNA integrity (Agilent Technologies, Santa Clara, CA, USA). Sequencing libraries were generated using the NEBNext^®^ UltraTM RNA Library Prep Kit for Illumina^®^ (New England Biolabs, Beijing, China) by following the manufacturer’s recommendations. The HiSeq X-TEN/NovaseqS4 PE Cluster Kit (Illumina, San Diego, CA, USA) was used to cluster different groups of samples on the cBot Cluster Generation System (Illumina, San Diego, CA, USA). Deferentially expressed mRNAs were selected according to Log2 (fold change) > 1 or *p*-value < 0.05 by DESeq2 R package (1.20.0). Gene Ontology (GO) enrichment analysis (adjusted *p*-value 0.05) and Kyoto Encyclopaedia of Genes and Genomes (KEGG) pathway analysis were used for functional annotation.

### 2.7. Statistical Analysis

GraphPad Prism 8 was used to conduct the statistical analysis. The data were displayed as means ± standard deviations. One-way ANOVA with Tukey’s multiple comparisons test was used to establish the statistically significant differences. A *p*-value of less than 0.05 was considered to indicate statistical significance.

## 3. Results

### 3.1. Screening of Inosine and Guanosine Degrading Lactic Acid Bacteria 

A total of 23 lactic acid bacteria strains were obtained using the selective culture of eight samples. Four tests were performed using these 23 bacteria strains. Lact-1 and Lact-2 were screened in the acid tolerance test (survival rates were 67.65 ± 0.98% and 76.17 ± 1.51%) and most strains performed well in the bile tolerance test (survival rate > 95%). Six strains of lactic acid bacteria grew significantly in the simulated gastric fluid environment, which were *Lactobacillus rhamnosus* RH01103, RH01232, Lact-1, *Animal Bifidobacterium subsp. lactis* RH04020, *Limosilactobacillus fermentum* RH08050, and Lact-2; the survival rate were 99.09 ± 0.61%, 96.94 ± 0.61%, 96.95 ± 1.38%, 99.07 ± 0.67%, 98 ± 0.92%, and 94.53 ± 0.55%. *Lactobacillus rhamnosus* RH01103, Lact-1, *Lactiplantibacillus plantarum* RH03010, *Animal Bifidobacterium subsp. lactis* RH04020, RH04568, and Lact-2 were screened in the artificial intestinal fluid (99.23 ± 0.37%, 98.21 ± 0.5%, 100 ± 0.12%, 100 ± 0.75%, 99.2 ± 0.69%, and 100 ± 0.21%). Overall, eight strains were screened out, showing excellent intestinal tolerance, as shown in Figure 1A and Appendix A.

The eight strains were tested for inosine and guanosine assimilating abilities based on HPLC detection. Compared with the control group without probiotics, Lact-1, Pediococcus acidilactici RH27102 showed the most significant decomposition ability of inosine and guanosine after 1 h co-culture (*p* < 0.001). After a 2 h culture with different lactic acid bacteria, *Limosilactobacillus fermentum* RH08050, Lact-1, and Lact-2 showed the most significant decomposing ability of inosine and guanosine (*p* < 0.001). Based on the performance of the purine nucleotide degradation ability of these strains of lactic acid bacteria, Lact-1 and Lact-2 were selected for the following animal tests in vivo (Figure 1B).

### 3.2. Lact-1 Treatment Alleviated Hyperuricemia in Mice 

The animal experiments were conducted as shown in Figure 2A. After 21 days, the serum UA levels increased significantly in potassium oxonate and adenine-induced hyperuricemia mice. It was found that the levels of serum UA decreased significantly after the oral administration of Lact-1 (*p* < 0.05) and the creatinine level decreased both in the Lact-1 and Lact-2 groups (Figure 3B). 

To evaluate the intestinal permeability of hyperuricemia mice post-intervention of the two selected probiotic strains, the levels of colon tight junction gene mRNA expression, i.e., Claudin-1, Claudin-4, Occludin, and Zo-1 of the hyperuricemia mice colon, were detected. In the model group, the expression levels of Claudin-1 and Claudin-4 were reduced significantly compared to the control group (*p* < 0.01, *p* < 0.05), and the mRNA expression of Occludin and Zo-1 had a downward trend. In contrast, after treatment with Lact-1 and Lact-2, the expression levels of the four genes increased, especially Lact-2 in Claudin-1 (*p* < 0.05) and Lact-1 in Occludin (*p* < 0.0001) (Figure 3C). It was observed that the expression of LPS in blood and liver was significantly induced in the hyperuricemia model group compared to the control while the LPS level in Lact-1 and Lact-2 was substantially downregulated compared to the model (Figure 3D). This may be due to increased permeability of the intestinal mucosa, prompting LPS to enter the bloodstream. The finding of this study indicates that the administration of Lact-1 and Lact-2 effectively improves intestinal permeability in hyperuricemia mice.

The signaling pathway involving TLR4, Myd88, and NF-κB is of critical importance in the regulation of the inflammatory response. The mRNA expression level of TLR4, Myd88, and NF-κB was substantially upregulated in the model group compared to the control, and Lact-1- and Lact-2-treated groups were downregulated compared to the model group, especially Lact-1 in Myd88 (*p* < 0.05) (Figure 3E). The level of pro-inflammatory cytokines in the hepatic level, i.e., IL-1β, IL-6, and IL-18, was upregulated in the model group; Lact-1- and Lact-2-treated groups significantly decreased the level of pro-inflammatory cytokines, especially IL-1β and IL-6 (Figure 3F). The hyperuricemia model group exhibited a substantial upregulation in the mRNA expression level of XOD. Nevertheless, the administration of Lact-1 and Lact-2 in animal subjects resulted in the restoration of XOD mRNA levels to their baseline values (Figure 3G).

### 3.3. Effect of Lact-1 and Lact-2 on Gut Microbiota of Hyperuricemia Mice

To investigate the effect of the two selected probiotics on the gut microbiota of hyperuricemia mice, we analyzed the change in fecal metagenomics through 16S rRNA gene sequencing. Alpha diversity analysis results showed that the Chao1 index of the model group was lower than the control group in the first, second, and third week, while it was also observed that the Lact-1- and Lact-2-treated group Chao 1 index was higher compared to the model group after the first, second and third week, but there was no statistical difference (Figure 3A). Principal component analysis (PCoA) was performed to identify variations in the fecal microbiome between the control, model, and treated groups (Lact-1 and Lact-2). Our data revealed that in Week 1, the model showed a distinct cluster separate from Lact-1, Lact-2, and the control group. In Week 2, the model formed a cluster different from the control group, indicating structural differences among the samples. Additionally, Lact-1 and Lact-2 were closely grouped compared to the model group. In Week 3, the control and Lact-1 groups clustered together, suggesting that the composition of species in these samples was more similar to each other compared to the model group (Figure 3B). Compared to the normal group, the Firmicutes/Bacteroidota (F/B) ratio of mice showed an increasing trend in the hyperuricemia model group, and the microbial imbalance occurred. An elevated F/B indicates a metabolic disorder that affects urea production and excretion [30]. After the administration of Lact-1 and Lact-2, the F/B ratio tended to normalize, and the dysbacteriosis was adjusted (Figure 3C–E).

We further assessed the fecal bacterial composition in the hyperuricemia model and different treated groups (Lact-1 and Lact-2) (Figure 3C). In the first week, at the phylum level, the control group had more Firmicutes but less Bacteroidota, Campilobacterota, Proteobacterota, and Actinobacteriota compared to the model group. In Lact-1, there were increased Firmicutes, Bacteroidota, and Desulfobacterota and a decrease in campylobacteria compared to the model group. Lact-2 had higher Actinobacteriota, Proteobacterota, Verrucomicrobiota, and Desulfobacterota than the model group. In the second week, the control group had higher Firmicutes and Actinobacteriota, while the model group had more Bacteroidota. The treated groups showed higher Bacteroidota and Actinobacteriota. In the third week, Firmicutes and Bacteroidota increased in the control group. Actinobacteriota, Proteobacteria, and Verrucomicrobiota increased in the model group. Lact-1 group had more Firmicutes and Proteobacteria, while Lact-2 had increased Bacteroidota and Proteobacteria. At the genus level, the hypouricemic group exhibited significant microbiota changes. In the first week, the control group had more *Lactobacillus*, while the model group had increased *Muribaculaceae*, *Helicobacter*, and *Lachnospiraceae NK4A136 group.* In week two, *Muribaculaceae*, *Bacteroides* in Lact-1, and *Alloprevotella* increased, while *Lactobacillus* and *Muribaculaceae* decreased in Lact-2. In week three, both Lact-1 and Lact-2 showed more *Lactobacillus* and *Muribaculaceae*, with Lact-1 having increased *Mitochondria* and Lact-2 having reduced *Bacteroides* compared to the model group (Figure 2D). The taxonomic distinctions among the groups were shown. In a comparison between the control gut microbiota, the model, and treated groups, the LEfSe analysis revealed that the control group prominently featured the genera *Lactobacillus* and *Odoribacter*, whereas *JG30KFCM45* and *Eubacterium xylanophilum group* significantly increased in Lact-1 and Lact-2, respectively. Additionally, *Helicobacter* exhibited a significant increase in week one, with non-significant differences observed among the groups in week two. During the third week, significant increases were observed in the control group, particularly in the Pantoea and Lact-1 groups, in which significant abundance was observed at the genus level, i.e., *Eubacterium xylanophilum group*, *Prevotellacaeae UCG 001*, and *Clostridia vadin BB60 group*. Additionally, the significant abundance of *Parbarteroides* and *Alistipes* in the Lact-2 group was notably higher. In contrast, in the model group, there were significant increases in *Lactobacillus* and *Domibacillus* (Figure 3H,I). 

### 3.4. Effect of Lact-1 and Lact-2 on UA Transporter in Small Intestine

The synthesis and excretion of UA are balanced in healthy people. The kidneys excrete 70% of the total UA from the human body, and the remaining 30% of UA is excreted through the small intestine [31]. The excretion of UA in the intestine is mainly dependent on UA transporters, which exist in human intestinal epithelial cells, and their production quantity, distribution range, and UA-lowering effect are affected by the gut microbiota.

ABCG2 and SLC2A9 play a role in the regulation of UA. The effect of Lact-1 and Lact-2 on mRNA expression of ABCG2 and SLC2A9 in the small intestine is shown in Figure 4A. The remarkable downregulation of ABCG2 and SLC2A9 was observed in the model group compared to the control group. In the treated group, the expression level of ABCG2 was significantly increased with respect to the model group, and the expression level of SLC2A9 was upregulated in the treated group with respect to the model group, but no statistical differences were observed. The results were further confirmed through Western blotting; the expression levels of ABCG2 and SLC2A9 decreased in the model group while the expression of the UA transporter was induced (Figure 4B,C). These findings represented that the treatment groups of Lact-1 and Lact-2 had remarkable effects on mRNA and protein on UA transporters genes ABCG2 and SLC2A9, especially Lact-1.

Gut microbiota can produce short-chain fatty acids (SCFAs), which can regulate the proliferation and repair of intestinal epithelial cells, thus affecting the expression of UA transporters. Our study found that Lact-1 and Lact-2 interventions increased the abundance of butyrate-producing bacteria, such as *Akkermansia*, *Bifidobacterium*, *Pseudomonas*, and *Lactobacillus* (Figure 4D).

ATP is the body’s energy source and supports active transport within cells. Previous studies have found that the gut of gout patients is rich in the bacterial metabolites acetate, succinate, and glucose, which provide energy for the intestinal epithelium to excrete UA through the transporter ABCG2 and SLC2A9 [32]. In our study, the levels of ATP were significantly reduced in the feces of the model group compared to the control group, whereas those in the Lact-1- and Lact-2-treated groups significantly increased (Figure 4F). A previous study showed that giving glycine and tryptophan supplements to gout patients accelerated the excretion of UA [33]. Gut microbiota like *Lactobacillus* species is known to produce proteolytic enzymes that help break down proteins into amino acids. The α-ketoacids produced by the decomposition of amino acids in the body can be oxidized to carbon dioxide and water through the tricarboxylic acid cycle and release energy. In the hyperuricemia mice, the levels of the amino acid were increased with respect to the control group, whereas the levels of the amino acid were substantially decreased in the Lact-1- and Lact-2-treated groups (Figure 4G). These findings suggest that Lact-1 and Lact-2 may convert amino acids into energy and promote UA excretion.

The enzymes produced by the gut microbiota also play a role in regulating the production and breakdown of UA. *E. coli* can secrete XDH, which converts purine into UA. In the metabolism of UA, *Lactobacillus*, *bifidobacteria*, and *pseudomonas* can produce allantoinase, which breaks down UA into urea and finally downregulates the UA levels in the serum [34]. In our study, the abundance of *Escherichia-Shigella* was higher in hyperuricemia mice, the oral administration of Lact-1 and Lact-2 increased the abundance of *Escherichia-Shigella*, especially Lact-1. The secretion level of XDH is basically positively correlated with the abundance of *Escherichia-Shigella*, which can convert the purine in hyperuricemia mice into UA (Figure 4E). The mRNA expression levels of fecal XDH were reduced in the model group compared to the control. On the contrary, the XDH expression levels in Lact-1 increased significantly (*p* < 0.05) (Figure 4H). Then, the abundance of *Lactobacillus*, *bifidobacterium*, and *Pseudomonas* decreased in the model group and increased in the Lact-1- and Lact-2-treated hyperuricemia groups (Figure 4D). These bacteria can secrete Allantoinase, and the expression level of allantoinase was positively correlated with the abundance of these bacteria in order to decompose UA into urea. The expression level of Allantoinase was downregulated compared to the control group, whereas in the Lact-1- and Lact-2-treated groups, the expression levels were upregulated with respect to the model group. This indicates that purines in food, with the help of gut microbiota, are first converted into UA and then broken down into urea and excreted from the body (Figure 4I).

### 3.5. Transcriptome Sequencing of Lact-1 on HT-29 Cells In Vitro

The results above indicate that Lact-1 and Lact-2 can attenuate hyperuricemia by regulating intestinal microbiota and UA transporter, especially Lact-1, which has a more significant effect on lowering serum UA. Whether the bacteria itself or the metabolites play the role of ameliorating hyperuricemia was further verified by transcriptomics, and we also searched for the possible mechanism of lowering UA.

Human Colon Cancer Cells (HT-29) are stimulated by Lact-1 and its metabolites; differential expression analyses of metabolites vs. control and probiotics vs. control were performed. The data showed that 1972 genes were upregulated and 496 genes were downregulated, whereas, in probiotics vs. control, 677 genes were upregulated while 483 genes were downregulated (Figure 5A,C).

#### 3.5.1. GO Functional Enrichment Analysis of Metabolites- and Probiotics-Treated HT-29 Cells

From the results obtained through GO enrichment analysis, we chose the most significant 30 terms to generate a histogram for graphic representation. If there were fewer than 30 terms, we included all of them in the histogram, as demonstrated in the figure below. To expansively investigate the impact of metabolites and probiotics on colon cell line HT-29, we used Omics share for conducting GO enrichment analysis. The results from the GO enrichment analysis of metabolites in the treatment of the Colon cell line HT-29 reveal their effects on several biological processes. Among these biological processes, the top three are extracellular structure organization, organic hydroxy compound metallic process, and extracellular matrix organization. The top three cellular components are the extracellular matrix, plasma membrane protein complex, and proteinaceous extracellular matrix. At the molecular functional level, ion channel complex, metal ion transmembrane transport activity, and ion gate channel activity were observed (Figure 5B). Probiotics act on HT-29 cell lines through various biological processes, and among the top three are response to hexose, response to monosaccharide, and regulation of vascular development; and the top three cellular components are MCM complex, keratin filament, and intermediate. At the molecular functional level, proximal promoter sequence-specific DNA binding filament, RNA polymerase 2 proximal promoter, and transcription factor activity RNA polymerase were observed (Figure 5D).

#### 3.5.2. KEGG Functional Enrichment Analysis of Metabolites and Probiotics

Significantly upregulated and downregulated metabolite and probiotics were subjected to enrichment analysis. The first enrichment analysis we conducted was with the KEGG database. The data revealed that upregulated metabolites were significantly enriched in the Ras signaling pathway, cAMP pathway, beta-Alanine metabolism, serotonergic synapse, Tyrosine metabolism, Aldosterone synthesis and secretion, cell adhesion molecules, Arachidonic acid metabolism, and cGMP-PKG pathway (Figure 5E). KEGG pathway analysis represented that metabolite downregulated various pathways i.e., cytokines–cytokines receptor interaction, glycine, serine and threonine metabolism, Alanine aspartate and glutamate metabolism, NOD-like receptor pathway, RIG-1-like receptor pathway, AGE-RAGE signaling pathway, IL-17 signaling pathway, glutathione metabolism, lipid and atherosclerosis, alcoholism, neutrophil extracellular trap formation pathways, and systemic lupus erythematosus were significantly enriched (Figure 5F).

The data showed that the upregulated probiotics were substantially enriched in insulin resistance, steroid hormones biosynthesis, protein digestion and absorption, phospholipases D signaling pathway, retinol metabolism, linoleic acid metabolism, AGE-RAGE signaling pathway, PI3K/Akt signaling pathway, MAPK signaling pathway, TNF signaling pathway, and C-type lectin receptor signaling pathway (Figure 5G). According to KEGG pathway analysis, probiotics downregulated the Wnt-signaling pathway, cytosolic DNA-sensing singling pathway, lipid and atherosclerosis, pyrimidine metabolism, hippo signaling pathway, base excision repair, vitamin digestion and absorption, nucleotide excision repair, protein processing in the endoplasmic reticulum, antigen processing and presentation, RNA polymerase, and steroid biosynthesis and DNA replication (Figure 5H). It has been reported that anti-hyperuricemia pathways include the cAMP pathway, TNF pathway, and MAPK, PI3K/AKT pathway [35,36], so we think that the bacteria and metabolites of Lact-1 work through different pathways, but the role of bacteria seems to be greater; specific mechanisms need to be followed up on.

## 4. Discussion

The prevalence of hyperuricemia is rapidly increasing throughout the world. According to recent studies, excessive levels of UA are a risk factor for gout, renal stones, heart disease, and stroke [37]. Therefore, it is crucial to find a safe and effective way to reduce UA, and probiotic therapy has received much attention in this field. In our study, the gastrointestinal tolerance of probiotics was first monitored to ensure their future colonization and effectiveness in the intestine. The attenuating effects of Lact-1 on hyperuricemia mice were evaluated. Our study lays a foundation for the future development of probiotic Lact-1 to alleviate hyperuricemia (Figure 6A).

Dairy products and human gut microbiota are important sources of probiotics. We isolate candidate strains from the gut microbiota of infants in these regions with the richest dairy products in China, as well as in the gut microbiota of infants in these regions and Bama, Guangxi, where life expectancy is longer. Effective probiotics have been isolated in these areas [38]. Most researchers have focused more on the ability of probiotics to degrade purine nucleosides in order to improve hyperuricemia efficacy, rather than their gastrointestinal tolerance [22]. Our study first identified the gastrointestinal tolerance of candidate strains; eight strains that have good gastrointestinal tolerance were screened. This screening process ensures that the screened strains can colonize the gut and play a beneficial role. Being able to colonize the gut is a prerequisite for being an effective probiotic.

Normalizing SUA levels is essential to reduce the risk of hyperuricemia. In our study, we revealed that Lact-1 can modulate the SUA level of PO+AD-induced hyperuricemia mice. After the intervention of Lact-1, the mRNA expression of tight junction protein was significantly increased, the content of LPS was decreased, the levels of NF-κB pathway and inflammatory factors were decreased, and XOD mRNA levels were also downregulated. In our study, we examined whether the administration of probiotics can upregulate the expression of the tight junction protein, improve and restore the dysbiosis of intestinal gut microbiota, and repair the intestinal impairment of the colon. Zo-1 and Occludin are integral proteins found in the tight junctions of the intestinal epithelium. These proteins are of utmost importance in maintaining the integrity of the intestinal mucosal barrier [39,40]. Similarly, a study was reported by Liang et al., where the administration of *L. gasseri* LG08 and *L. mesenteroides* LM58 orally was found to have a notable impact on the expression of Occludin and Claudin proteins. These proteins are known to have a crucial role in preserving the integrity and functionality of the intestinal mucosal barrier [41]. Furthermore, the study determined that sodium butyrate had a potential therapeutic effect and impaired intestinal mucosa barrier induced the mRNA and protein expression levels of mus-2, Zo-1, Occludin, and ABCG2, which reduced UA production [42]. After Lact-1 and Lact-2 administration, along with the repair of the long mucosal barrier, LPS levels in both serum and liver were reduced. The loss of the intestinal barrier caused by dysbiosis of gut microbiota leads to the translocation of intestinal-derived LPS, which plays an important role in the development of gout caused by hyperuricemia [43]. The changes in gut microbiota can decrease intestinal permeability, leading to changes in the entry of LPS into the circulatory system, thereby controlling metabolic endotoxemia, inflammation, and related diseases [44]. Previous studies reported that hyperuricemia animals exhibit heightened levels of pro-inflammatory dysbiosis in intestinal microbiota. This dysbiosis gives rise to decreased intestinal permeability and elevated serum LPS levels. LPS can activate the NF-κB pathway and keep the body in an inflammatory state. In our study, we observed that Lact-1 and Lact-2 treatment exhibited a more pronounced impact in inhibiting the activation of the TLR4/Myd88/NF-κB signaling pathway in the liver. On the other hand, the inflammation was studied in more detail in the kidney of the hyperuricemia mice model. Probiotics and prebiotics were reported to reduce the levels of UA in hyperuricemia, regulating UA excretion and transportation, thereby inhibiting renal inflammation and inhibiting the TLR4/Myd88/NF-κB signaling pathway [45,46,47,48]. Our results indicated that hepatic pro-inflammatory markers were reduced, i.e., IL-1β, IL-6, and IL-18. A similar study observed that *L. plantarum* Q7downregulated the levels of IL-1β, TNF-α, and MDA in hepatic and nephritic homogenate, indicating improved antioxidant activity and relief inflammation symptoms in the treatment of hyperuricemia mice [35]. The polyphenols extracted from purple potato leaves inhibited the levels of IL-1β, IL-6, and TNF-α, as well as serum creatinine, and also decreased the levels of expression of XOD in the liver of hyperuricemia mice, thus protecting the kidney [49]. As is well-known, purine produces UA under the action of XOD, which is the rate-limiting enzyme of UA production. In our study, we reported that probiotics inhibited the expression of XOD in the liver, which plays a key role in the production of UA and lowers UA levels. A similar study noted that *L. paracasei MJM60396* decreased purine absorption and lowered the UA concentration in serum, inhibited expression of XOD, restored the intestinal barrier, and upregulated the levels of Zo-1 and Occludin in hyperuricemia mice [23]. There is a lot of evidence that prebiotics and probiotics improve intestinal barrier function. We found that hyperuricemia mice had decreased intestinal barrier function, which is consistent with some earlier studies [50]. The restoration of intestinal wall function may, as previously mentioned [50,51,52], result in lower serum LPS concentrations, reduced systemic inflammation, decreased XOD activity, and inhibition of UA formation. 

The 16S rRNA gene-sequencing results showed that the oral administration of Lact-1 can increase the abundance of gut microbiota in hyperuricemia mice, thus making its microbiota structure closer to that of normal mice. Patients with hyperuricemia always have an imbalance of intestinal microbiota, and restoring the balance of intestinal microecology can reduce the symptoms of hyperuricemia. It is widely believed that the F/B ratio has an important influence on maintaining normal intestinal homeostasis [53]. The increase or decrease in the F/B ratio is the manifestation of biological dysregulation. In our study, the oral administration of Lact-1 regulated the gut microbiota and decreased the higher F/B ratio in hyperuricemia mice to normal; these results indicated that Lact-1 could regulate the intestinal microbes of hyperuricemia mice.

Lact-1 reduced the expression of XOD and the production of UA by regulating the balance of intestinal microecology, and our study also focused on UA excretion. The oral administration of Lact-1 enhanced the protein expression level of ABCG2 while increasing the abundance of short-chain fatty acid-producing gut microbiota. The levels of ATP, XDH, and allantoinase in feces were elevated; at the same time, the amino acid content decreased. ABCG2 and SLC2A9 genes are involved in the regulation of UA in the body. When genetic mutations or other factors disrupt the normal function of these genes, it can lead to hyperuricemia, which, in turn, can result in health issues such as gout and other related conditions. In our study, the expression of UA transporters ABCG2 and SLC2A9 increased in probiotics (Lact-1 and Lact-2)-treated groups. Probiotics and prebiotics have been shown to ameliorate hyperuricemia in mice by modulating the gut microbiota and increasing the expression of UA exporter genes such as ABCG2 and SLC2A9 [54,55]. Furthermore, a study examined that *L. rhamnosus* Fmb14 cell-free-extract treatment ameliorates hyperuricemia through the inhibition of XOD and increases the expression of ABCG2 UA transporter [56]. Overall, the SLC family, particularly SLC22A12 and SLC2A9 along with ABCG2, play important roles in UA transportation, and their dysfunction can contribute to hyperuricemia and gout [57]. The gut microbiota also helps excrete exogenous UA. It enhances UA-gut microbiota interaction, and intestinal microbiota dysbiosis affects UA excretion. SCFAs can regulate the formation and repair of intestinal epithelial cells, and the increased abundance of bacteria that produce SCFAs contributes to UA excretion. In our study, *Akkermansia*, *Bifidobacterium*, *Pseudomonas*, and *Lactobacillus* that can produce SCFAs increased after Lact-1 and Lact-2 treatment. The research noted that camellia japonica bee polyphenols restored the gut microbiota of the hyperuricemia model, induced the abundance of Lactobacillus, and increased the contents of SCFAs [58]. Similarly, according to LEfSe analysis in a report, green alga Enteromorpha prolifera polysaccharide-treated hyperuricemia models showed an increase in the level of Alistipes and Parasutterella [59]. Enteromorpha clathrate polysaccharides increased butyrate-producing Eubacterium xylanophilum and alleviated high-fat diet-induced intestinal dysbiosis [59]. Furthermore, chlorogenic acid treatment increased short-chain fatty acid bacteria, especially Prevotellacaeae UCG 001 in hyperuricemia mice [46]. Berberine improved gut microbiota and increased beneficial bacteria such *Lactobacillus*, *Alistipes*, *Prevotellaceae UCG 001*, and *Parasutterella*, restoring intestinal homeostasis in hyperuricemia mice [60]. In our study, the treated groups had higher alpha diversity and Bacteroidota, Firmicutes, and Proteobacteria than the hyperuricemia model group. At the genus level, *Lactobacillus* was abundant, with Lact-1 having more Mitochondria and Lact-2 having fewer Bacteroides than the model group. We also found taxonomic differences between the groups, with Lact-1 having *Eubacterium xylanophilum group*, *Prevotellacaeae UCG 001*, and *Clostridia vadin BB60* group and Lact-2 having *Parbarteroides* and *Alistipes*. Many genera that increased in the hyperuricemia model may be responsible for excessive UA levels. Feng et al. reported that β-carotin and green tea powder restore the abundance of Bacteroidetes at the phylum level and genus level, decreasing the relative abundance of *Lachnospiraceae NK4A136 group* and *Ruminococcaceae UCG-014* in the model group [61]. Furthermore, it was reported that tuna meat oligopeptide induced *Bacteroides*, *Bacilli*, and *Actinobacteria* and reduced *Epsilonproteobacteria* in hyperuricemia [45]. The model group with hyperuricemia had lower ATP levels than the control and probiotics groups for multiple reasons; for example, metabolic disturbances, oxidative stress, inflammation, and gut dysbiosis can impact ATP synthesis and utilization in hyperuricemia [62,63]. ATP provides energy for UA to be excreted from the body via UA transporters. Probiotics like *Lactobacillus* strains may increase gut microbiota and the levels of ATP [41]. ATP generation is also affected by nutrition, notably high-purine diets [34]. The gastrointestinal system breaks down food proteins into amino acids using proteolytic enzymes from *Lactobacillus* species [64]. An increased abundance of *Lactobacillus* may raise amino acid levels in feces, indicating that they use dietary proteins more efficiently [65]. Increased gut amino acid levels indicate better protein utilization, nutritional absorption, and gastrointestinal health [66]. But our study showed that there were higher levels of amino acid in hyperuricemia mice and lower levels after treatment with Lact-1 and Lact-2. The probable reason may be that amino acids in the body can be further decomposed and converted into sugars, lipids, or some non-essential amino acids, and they can also be oxidized into carbon dioxide and water through the tricarboxylic acid cycle and release energy. Amino acid in *Lactobacillus*-treated groups transforms into energy to assist UA excretion. Another possibility is that amino acid metabolism contributes less to the excretion of UA through the gut.

GO functional enrichment analysis showed that Lact-1 had the greatest effect on extracellular structure organization, organic hydroxy compound metallic process, and extracellular matrix organization; Lact-1 affected the plasma membrane protein complex. At the molecular functional level, the ion channel complex, metal ion transmembrane transport activity, and ion gate channel activity were affected. KEGG functional enrichment analysis showed that the probiotics upregulated genes were substantially enriched in the PI3K/Akt signaling pathway, MAPK signaling pathway, and cAMP pathway. Previous studies indicate that AKT has an important role in hyperuricemia and gout. Activation of the AMPK/AKT/CREB pathway boosts CREB’s binding to the ABCG2 promoter, elevating ABCG2 expression and UA excretion [67]. The upregulation of β-Alanine is directly linked with hyperuricemia, but it was reported that it improves the intestinal morphology and gut barrier function, which may reduce hyperuricemia [68]. Additionally, a study reported that under hyperuricemia conditions, the Tyrosine metabolism was downregulated [69]. Mehmood et al. found that grapefruit lowered UA levels, reduced inflammation markers, enhanced ABCG2 UA transporter expression, and improved renal and intestinal health in the hyperuricemia mice model [70]. In our study, we found that metabolites treated in colon cells were significantly upregulated in the cAMP pathway, and the activation of the cAMP pathway can enhance cell activity and inhibit inflammatory responses. Furthermore, a few studies reported that Chlorogenic acid has been found to reduce UA, Cr, BUN, AST, ALT, and LPS levels, inhibit XO, decrease mRNA expressions of IL-1β, TNF-α, and NOD-like receptors, promote mRNA expressions of tight junction genes in the colon, and inhibit the expressions of TLR4, MyD88, and NF-κB signaling pathways in the kidneys of hyperuricemia mice [46]. It was fond that α-Viniferin significantly lowered UA levels and protected the kidney against hyperuricemia-induced damage. It achieved this by inhibiting XOD to reduce UA formation, acting as a dual inhibitor to decrease UA absorption, and promoting the expression of UA exporters, ABGC2 and OAT1. Additionally, α-Viniferin downregulated pathways including IL-17, chemokine signaling, and PI3K/Akt [71]. Another study observed that elevated alcohol consumption increased the risk of hyperuricemia [72]. It was previously reported cytokines–cytokines receptor interaction and glycine, serine, and threonine metabolism were upregulated in hyperuricemia. It was also reported that systemic lupus erythematosus is directly linkage with hyperuricemia [73,74,75]; however, in our study, we noted that in metabolites-treated colon cells, these pathways were downregulated. The Gegen Qinlian Decoction (GGQLD) can effectively treat asymptomatic hyperuricemia patients by modulating the expression of circadian clock genes and regulating the proportion of innate lymphoid cells (ILCs) [76]. Conjugated linoleic acid reduced hyperuricemia, induced insulin resistance, and reduced renal inflammation in rats [77]. Qi et al. documented that PEW and LLW protein peptides lowered UA levels, suppressed ADA activity, inhibited hepatic and serum XOD, enhanced the expression of GLUT9, URAT1, ABCG2, OAT1, and OAT3, protected against kidney damage, and improved gut microbiota in a mouse model of hyperuricemia [78]. The oral administration of *L. pentosus* P2020 reduced UA, alleviated renal inflammation via the downregulation of NF-κB, MAPK, and TNF-α expressions, and upregulated UA transporters ABCG2 and SLC2A6 in hypouricemic mice [79]. In our study, we observed upregulation in the circadian rhythm, linoleic acid metabolism, protein digestion and absorption, as well as the ABC transporter pathways. Furthermore, probiotics downregulated various pathways, and among these pathways, some are directly linked with hyperuricemia. Lee et al. observed the link between excessively high UA levels and adverse outcomes in the mouse model, including elevated cholesterol and lipid peroxidation, both of which contribute to cardiovascular issues and atherosclerosis [80]. It was found that α-Viniferin significantly lowered UA levels and protected the kidney against hyperuricemia-induced damage. It achieved this by inhibiting XOD to reduce UA formation, acting as a dual inhibitor to decrease UA absorption, and promoting the expression of UA exporters ABGC2 and OAT1. Additionally, α-Viniferin downregulated pathways including IL-17, chemokine signaling, and PI3K/Akt [71]. Another study observed that elevated alcohol consumption increased the risk of hyperuricemia [72]. Elevated serum UA levels may exacerbate inflammation and kidney damage in systemic lupus erythematosus (SLE) patients [75]. The study found that API reduces UA, improves kidney health, and reduces fibrosis in hyperuricemia mice by enhancing urate excretion and downregulating the Wnt/β-catenin pathway [81]. In our investigation, the RNA sequencing data unveiled notable downregulation of specific pathways in probiotic-treated HT-29 cells. These pathways encompassed lipid and atherosclerosis, alcoholism, IL-17, PI3K/Akt, systemic lupus erythematosus, and the Wnt signaling pathway. 

These enhanced expression pathways are almost all through enhancing the intestinal mucosal barrier, reducing the entry of LPS into the blood circulation, inhibiting TLR-4/Myd88/NF-κB signaling pathway, reducing the release of inflammatory factors, thereby reducing the release of the rate-limiting enzyme of UA synthesis, xanthine oxidase, and thus reducing the synthesis of UA in the body. On the other hand, these pathways also enhance the excretion of UA in the intestine by increasing the expression of ABCG2 protein. By taking Lact-1 orally, it can adjust the imbalance of microbiota and increase the abundance of some beneficial bacteria, which can play a role in nourishing intestinal epithelium and enhancing protein expressions (Figure 6B).

## 5. Conclusions

In conclusion, *Limosilactobacillus reuteri* HCS02-001 sourced from traditional Chinese fermented milk such as curd has a strong gastrointestinal tolerance and good potential to become a probiotic, it reduces the synthesis of UA and enhances its excretion in the gut of hyperuricemia mice through the manipulation of PO+AD-induced intestinal dysbiosis. *Limosilactobacillus reuteri* HCS02-001 restored the gut microbiota structure, increased the abundance of beneficial bacteria, promoted cell survival through the PI3K/AKT pathway, nourished intestinal epithelium, increased the expression level of the intestinal tight junction gene, enhanced intestinal mucosal barrier, inhibited the TLR-4/Myd88/NF-κB pathway, and reduced the release of inflammatory factors. The cAMP pathway participated in cell proliferation and inflammation, it also enhanced the function of the intestinal mucosal barrier and inhibited the inflammatory response. Thus, the content of xanthine oxidase, the rate-limiting enzyme of UA synthesis, was reduced, and the synthesis of UA in PO- and AD-induced hyperuricemia mice was reduced. On the other hand, the metabolites of these beneficial bacteria can enhance the expression of UA transporters like ABCG2 and promote the excretion of UA in the intestine through the MAPK pathway. Taken together, our results demonstrate that *Limosilactobacillus reuteri* HCS02-001 can be used as a preventing and ameliorating probiotic for hyperuricemia. Our research has some limitations, for example, probiotics were collected from the feces of infants in longevity areas, but the gut microbiota of long-lived elderly people in longevity villages in China may also be an excellent source of probiotics. Thus, we can further expand the sources of probiotic screening. And our study did not provide a comparative analysis of *Limosilactobacillus reuteri* HCS02-001 and full-dose drugs. Further research is needed to determine whether probiotics are more cost-effective compared to some medications. 

## Figures and Tables

**Figure 1 microorganisms-12-00637-f001:**
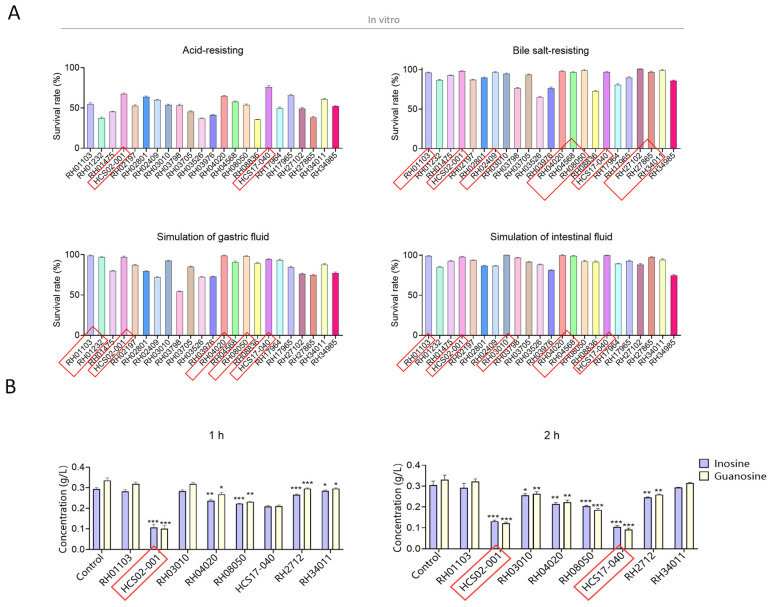
Screening lactic acid bacteria in vitro. (**A**) Tolerance to a simulated gastrointestinal environment of the 23 strains. (**B**) The content of inosine and guanosine after 1 h and 2 h culture with different probiotics. One-way ANOVA Tukey’s multiple comparisons test was performed. * *p* < 0.05, ** *p* < 0.01, and *** *p* < 0.001. The red boxes indicate the strains that performed well in this test.

**Figure 2 microorganisms-12-00637-f002:**
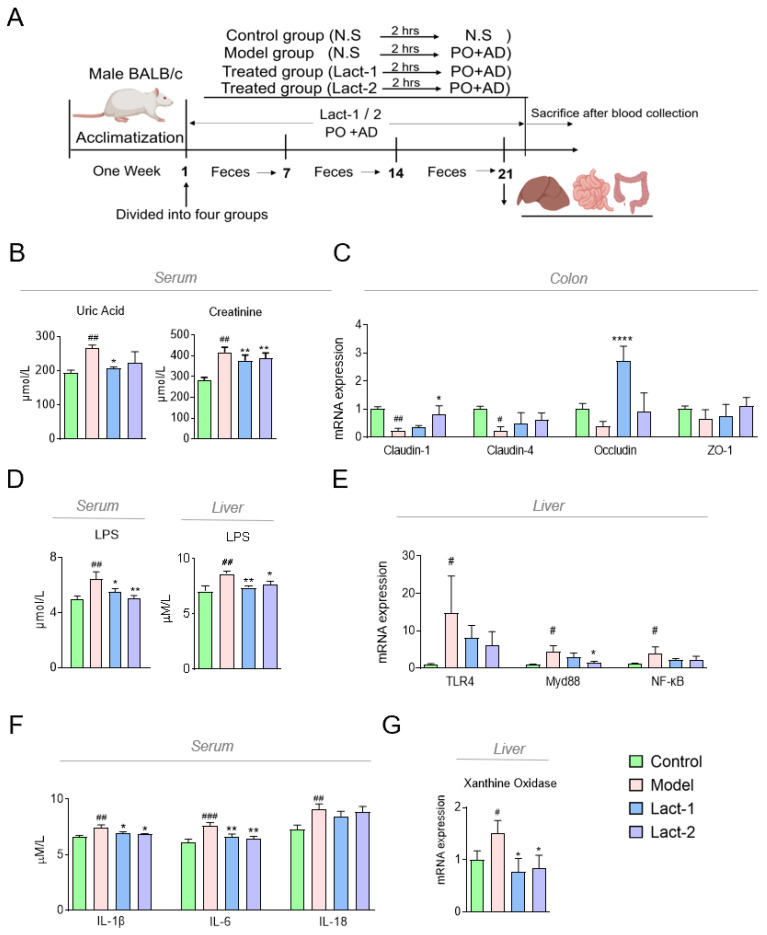
The pattern diagram of in vivo experiment. (**A**) Schematic representation of the experimental schedule. Control group, hyperuricemia model group with 200 mg/kg potassium oxonate (PO) + 50 mg/kg adenine (AD). Lact-1-treated group of hyperuricemia mice with 200 mg/kg potassium oxonate (PO) + 50 mg/kg adenine (AD), and Lact-2-treated group of hyperuricemia mice with 200 mg/kg potassium oxonate (PO) + 50 mg/kg adenine (AD). The control and hyperuricemia groups were given 200 μL normal saline, while the treated groups were orally administrated 200 μL of Lact-1 and Lact-2 (1×10^9^ CFU/mL). After 2 h, except for the normal group, the other groups were given the modeling drug for 21 days. Lact-1 and Lact-2 can reduce hyperuricemia-associated pro-inflammatory pathways and factors. (**B**) The levels of UA and creatinine in serum. (**C**) The mRNA expression of the tight junction protein of the colon. (**D**) The content of LPS in the serum and liver. (**E**) The mRNA expression of TLR4, Myd88, and NF-κB in the liver. (**F**) Serum IL-1β, IL-6, and IL-18 measurements in the serum. (**G**) The mRNA expression of Xanthine Oxidase in the liver. The ‘#’ symbol shows comparisons between the control and model groups, with significance levels represented as follows: # *p* < 0.05, ## *p* < 0.01, and ### *p* < 0.001. Additionally, significance levels for comparisons with other groups compared to the model group were indicated as * *p* <0.05, ** *p* < 0.01, and **** *p* < 0.0001.

**Figure 3 microorganisms-12-00637-f003:**
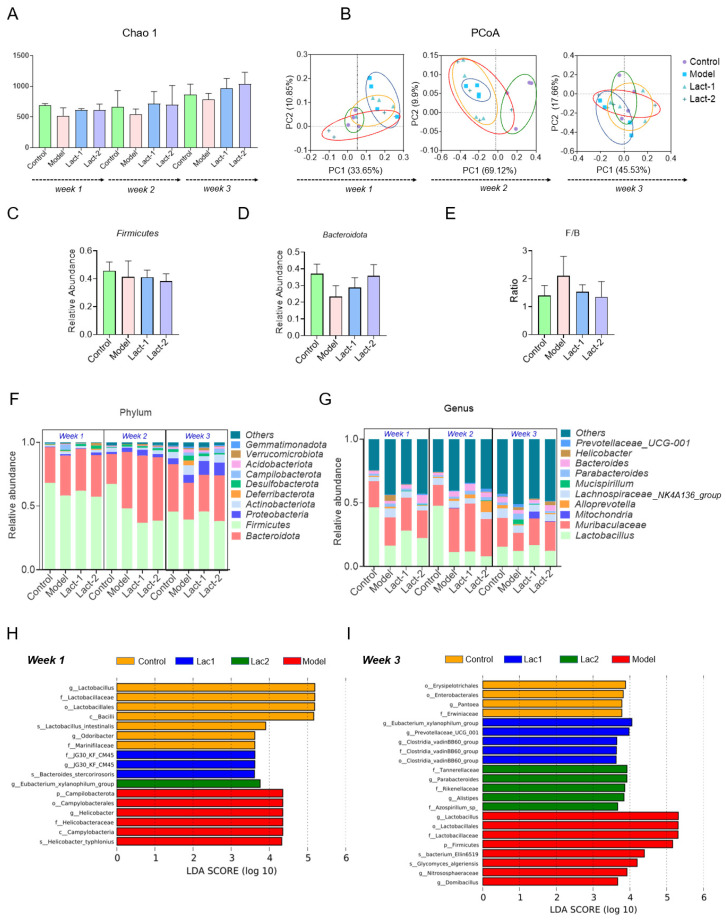
Effect of Lact-1 and Lact-2 on remodeling the gut microbiota. (**A**) The Chao1 index showing alpha diversity. (**B**) Principal component analysis (PCoA) plot showing the beta diversity. (**C**) The relative abundance of Firmicutes at the phylum level. (**D**) The relative abundance of Bacteroidota at the level of phylum. (**E**) The ratio of the relative abundances of Firmicutes and Bacteroidota (F/B). (**F**) Abundance distribution at the phylum level. (**G**) Abundance distribution at the genus level. (**H**) LEfSe analysis identified the most differentially abundant taxons. LDA scores of >4 are shown after the first week. (**I**) LEfSe analysis identified the most differentially abundant taxons. LDA scores of >4 are shown after the third week.

**Figure 4 microorganisms-12-00637-f004:**
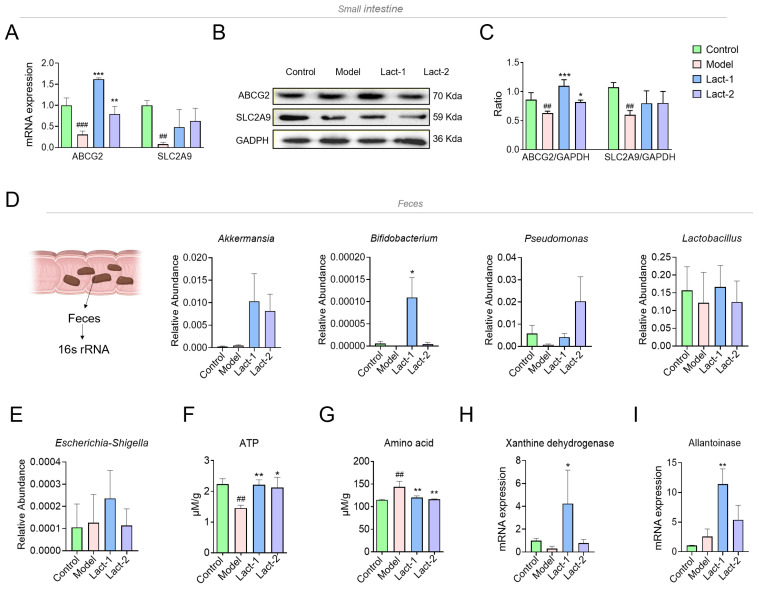
Effect of Lact-1 and Lact-2 on UA transporter and gut microbiota derivatives. (**A**) The mRNA expression of ABCG2 and SLC2A9 in the small intestine. (**B**) The expression and phosphorylation levels of proteins involved in the ABCG2 and SLC2A9 pathways were determined using Western blot analysis. GAPDH expression was detected as an internal control. (**C**) The mRNA expression of ABCG2/GADPH and SLC2A9/GADPH. (**D**) The relative abundance of *Akkermansia*, *Bifidobacterium*, *Pseudomonas*, and *Lactobacillus* in feces. (**E**) The relative abundance of *Escherichia-Shigella* in feces. (**F**) Fecal ATP level. (**G**) Fecal amino acid level and data were analyzed using the colorimetric method. (**H**) The mRNA expression of Xanthine dehydrogenase. (**I**) The mRNA expression of Allantoinase. The ‘#’ symbol shows comparisons between the control and model groups; ## *p* < 0.01, and ### *p* < 0.001. Significance levels for comparisons with other groups compared to the model group were indicated as * *p* < 0.05, ** *p* < 0.01, and *** *p* < 0.001.

**Figure 5 microorganisms-12-00637-f005:**
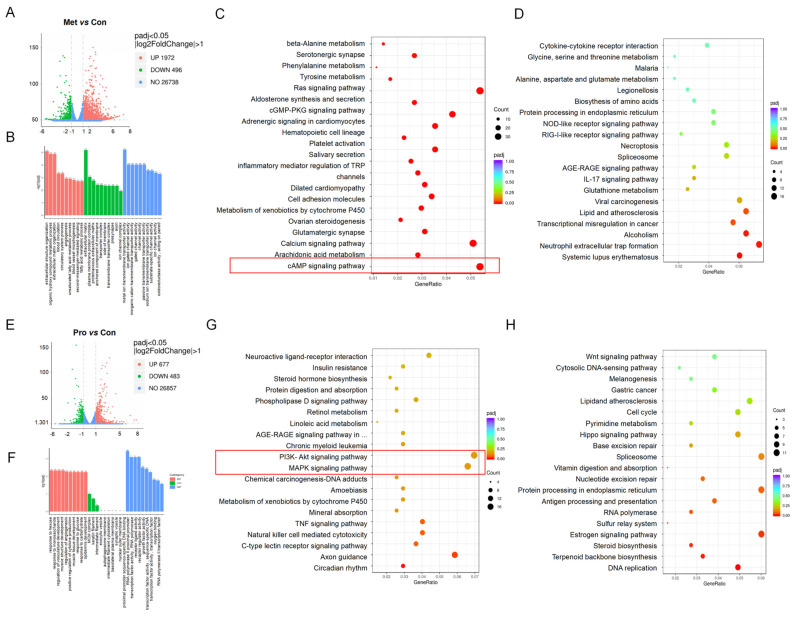
Effect of Lact-1 on HT-29 cells in vitro. (**A**) Volcano plot illustrating the differential gene expression (DEG) analysis between HT-29 cells treated with metabolites and normal HT-29 cells. (**B**) GO enrichment analysis on upregulated and downregulated DEGs in HT-29 cells treated with metabolites. Biological processes are denoted in red, cellular components in green, and molecular functions in blue. (**C**) Volcano plot analysis between HT-29 cells treated with probiotics and normal HT-29 cells. (**D**) GO enrichment analysis on upregulated and downregulated DEGs in HT-29 cells treated with probiotics. (**E**) Bubble chart illustrating the enrichment analysis of upregulated KEGG pathways in HT-29 cell lines treated with metabolites. (**F**) Bubble chart displaying the enrichment analysis of downregulated KEGG pathways in HT-29 cell lines treated with metabolites. (**G**) Bubble chart depicting the enrichment analysis of upregulated KEGG pathways in HT-29 cell lines treated with probiotics. (**H**) Bubble chart demonstrating the enrichment analysis of upregulated KEGG pathways in HT-29 cell lines treated with probiotics. In the bubble charts, the size and color of bubbles represent the significance and magnitude of pathway enrichments, respectively.

**Figure 6 microorganisms-12-00637-f006:**
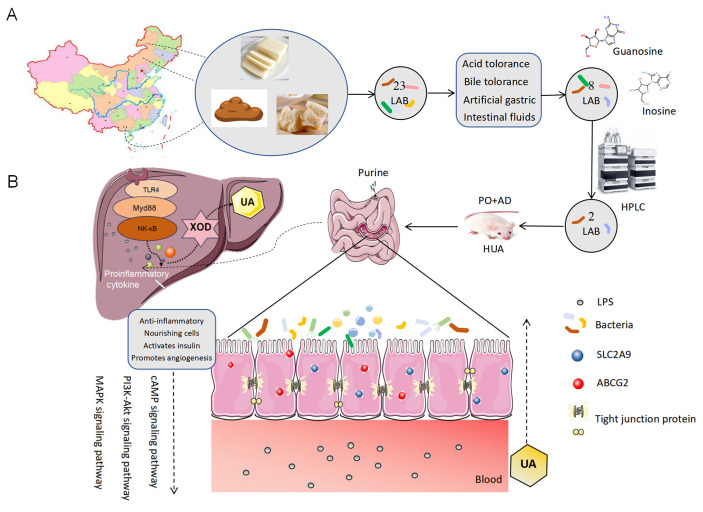
*Limosilactobacillus reuteri* HCS02-001 attenuates hyperuricemia through gut microbiota-dependent regulation of UA biosynthesis and excretion. In total, 23 lactic acid bacteria strains were isolated from the feces of healthy Chinese infants, and traditional fermented foods such as milk curd and lump milk were sourced from diverse regions in Inner Mongolia and Guangxi. Through in vitro gastrointestinal tolerance tests and purine degradation tests, 2 candidate strains were screened and administered orally to PO+AD-induced hyperuricemia mice. We found that Lact-1 had a better effect. Lact-1 repaired the dysbiosis of the gut microbiota in hyperuricemia mice, restored the intestinal mucosa, reduced LPS entering the bloodstream, and further reduced the release of inflammatory factors through the cAMP, MAPK, PI3K/AKT signaling pathways, thereby reducing the release of the rate-limiting enzyme XOD produced by UA, finally reducing UA production. On the other hand, Lact-1 increases the content of beneficial bacteria and the expression of UA transporters. (**A**) The process of screening probiotics in vitro. (**B**) Mechanism of *Limosilactobacillus reuteri* HCS02-001 attenuates hyperuricemia.

**Table 1 microorganisms-12-00637-t001:** Sampling spot and source.

Sample Number	Sampling Spot	Strain Source
1	Bayin Irangacha, Ewenke Banner, Hulunbuir, Inner Mongolia	Feces of healthy infant
2	Tongda Ranch in Manzhouli, Hulunbuir, Inner Mongolia	Fermented milk curd
3	West Ujur Sumu, Chenbarhu Banner, Hulunbuir, Inner Mongolia	Feces of healthy infant
4	Xilinhot, Xilingol League, Inner Mongolia	Feces of healthy infant
5	Bama Longevity Village, Guangxi	Feces of healthy infant
6	Dashimo, New Baerhuyou Banner, Hulunbuir, Inner Mongolia	Feces of healthy infant
7	Sanhe Hui Township, Erguna city, Hulunbuir, Inner Mongolia	Fermented milk lump
8	Chenbarhu Banner, Hulunbuir, Inner Mongolia	Fermented milk lump

**Table 2 microorganisms-12-00637-t002:** Target gene primer sequences.

Gene	Forward Primer (5′-3′)	Reverse Primer (5′-3′)
XOD	ATGACGAGGACAACGGTAGAT	TCATACTTGGAGATCATCACGGT
TLR4	CCTTTTCATCTCTGCCTTCACTAC	GGGACACCACGACAATAACCT
MyD88	GATGGTAGCGTTGTCTCTGAT	GATGCTGGGGAACTCTTTCTTC
NF-κB	CCCATGTAGACAGCACCACCTATGAT	ACAGAGGCTCAAAGTTCTCCACCA
Claudin-1	CCAGGTACGAATTTGGTCAGG	TGGTGTTGGGTAAGAGGTTGT
Claudin-4	TGGGGCTACAGGTAATGGG	GGTCTGCGAGGTGACAATGTT
Occludin	CTTCCAATGGCAAAGTGAATG	TACCACCGCTGCTGTAACGAG
ZO-1	GAGCCTAATCTGACCTATGAACC	TGAGGACTCGTATCTGTATGTGG
ABCG2	AAATGCTGTTCAGGTTATGTGGT	TCCGACCTTAGAATCTGCTACTT
SLC2A9	TTGCTTTAGCTTCCCTGATGTG	GAGAGGTTGTACCCGTAGAGG
XHD	CATGGGGAAGACAACCACAGG	ATGGTCCTGATCCTGGCATCC
Allantoinase	GCTTACGATATGGTGATA	TCAATGTCAGAACCTATC

## Data Availability

Data are contained within the article.

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
