# Peer review of "Limosilactobacillus reuteri HCS02-001 Attenuates Hyperuricemia through Gut Microbiota-Dependent Regulation of Uric Acid Biosynthesis and Excretion"

_microorganisms, 2024, doi:10.3390/microorganisms12040637_

Round 1

Reviewer 1 Report

Comments and Suggestions for Authors

Comments to the authors:

The introduction effectively outlines the study, emphasizing the assessment of probiotics in areas with abundant dairy products and longevity populations in China. The focus on reducing uric acid (UA) levels and exploring underlying mechanisms is well-established.

The selection of lactic acid bacteria from diverse sources, including healthy Chinese infant feces, traditional fermented foods (hurood), and lump milk, reflects a comprehensive approach. However, a brief mention of the rationale behind choosing these specific sources and their relevance to the study would enhance clarity.

The evaluation of candidate strains for probiotic potential based on their tolerance to acid, bile, artificial gastric juice, and artificial intestinal juice is a critical aspect. Identifying eight strains with superior tolerance to simulated intestinal conditions is a noteworthy finding that should be emphasized in the results section.

While the manuscript is well-written, there are small raised concerns about its innovation, and reader interest. Providing clarification or expanding on specific aspects of the study that could enhance uniqueness and significance may address these concerns.

The conclusions section requires more detailed elaboration. Present a concise summary of key findings and their implications. Discuss potential limitations of the study and suggest directions for future research.

Request a thorough review of the English language and grammar in the manuscript. Clarity and precision in language are crucial for effective communication.

Verify and standardize the reference style throughout the manuscript. Consistency in citation format is essential for maintaining professionalism and adhering to journal guidelines.

Overall, commendable methodology and results are noted in the study. Addressing concerns about originality, providing depth in the conclusions section, and implementing suggested improvements related to figures, language, and references will enhance the overall quality of the manuscript.

Comments on the Quality of English Language

Minor editing of English language required

Author Response

Dear reviewer,

I would like to express my deep gratitude to you for the constructive comments and suggestions.

A Point-by-Point Response to reviewers’ comments is given in the attachment.

Reviewer 2 Report

Comments and Suggestions for Authors

Interesting results. Please revise and correct according to the attached file.

Author Response

Dear reviewer

I would like to express my deep gratitude to you for the constructive comments and suggestions.

A Point-by-Point Response to reviewers’ comments is given in the attachment.

Reviewer 3 Report

Comments and Suggestions for Authors

To authors:

The present work is a very important contribution.

The results are relevant and the conclusions that can be taken are interesting.

However, there must be an improvement in the description of methodology. For example, there is no indiocation for the biomarkers to be analysed (this information appears only in the results section; HPLC method used is not described and there is no clue for what was measuered.

There is no refference to any ethical approvement, namely for the samples taken from children. It must be presented.

Figures are confused and information is missing (legends and axis). For example, in figure 1 there is no refference to the control used.

English must be improved and verbal/grammatical faults must be corrected.

Some bibliographic refferences could be improved (some have more than 10 yrs).

Comments on the Quality of English Language

English must be improved and verbal/grammatical faults must be corrected.

Author Response

(The authors gave the same response as above.)

Reviewer 4 Report

Comments and Suggestions for Authors

Well-written manuscript and project design. A few minor edits:

1. Affiliation - Needs to be written in a full form including the state and country.

2. Section 2.3. heading needs to be rewritten. 

3. Discussion can start with a general paragraph, followed by subsections describing each group of results.

Author Response

(The authors gave the same response as above.)
